# 3D Media Stabilizes Membrane and Prolongs Lifespan of Defolliculated *Xenopus laevis* Oocytes

**DOI:** 10.3390/membranes12080754

**Published:** 2022-07-31

**Authors:** Nikhil D. Aggarwal, Steven L. Zeng, Ryan J. Lashgari, Leland C. Sudlow, Mikhail Y. Berezin

**Affiliations:** Mallinckrodt Institute of Radiology, Washington University School of Medicine in S. Louis, 4515 McKinley, St. Louis, MO 63110, USA; aggarwalnikhil34@gmail.com (N.D.A.); steven.zeng@duke.edu (S.L.Z.); ryan.lashgari@wustl.edu (R.J.L.); lcsudlow@wustl.edu (L.C.S.)

**Keywords:** oocytes, *Xenopus laevis*, Cultrex, vitelline membrane, lifespan

## Abstract

*Xenopus laevis* oocytes are commonly used in many fundamental biological studies. One of the major limitations of *X. laevis* oocytes is their short storage lifespan with most defolliculated oocytes physically deteriorating in 10 days or less. Herein, we identified a 3D Cultrex-based storage media that incorporates extracellular membrane-based hydrogels to maintain oocyte integrity. Under these treatments, the lifespan of the oocytes increased to more than 20 days compared to standard conditions. The treatment preserved the oocytes membrane integrity and did not interfere with mRNA- or cDNA-derived protein expression.

## 1. Introduction

The oocytes of the African clawed frog, *Xenopus laevis,* have provided biologists with a useful tool in the field of biomedical research for over 50 years. *X. laevis* oocytes with their large physical size, which is approximately 1.2 mm in diameter, give researchers the opportunity to assess subcellular biological functioning on a macro scale and aid in the understanding of the egg/oocyte’s life cycle of mammals [1]. With regards to the subcellular machinery of *X. laevis* oocytes, the ability of the oocyte to express foreign DNA and RNA at a high success rate within a 48 -hour period is unique among other cells [2]. By utilizing the oocyte’s robustness and tolerance to physical manipulations, breakthroughs have been made in a multitude of fields, including electrophysiology [3], ion channel biology [4], drug discovery [5], and understanding the mechanisms of neurological diseases [6].

Despite the obvious benefits, *X. laevis* oocytes also possess certain serious disadvantages. One of the most hampering issues is that the typically utilized defolliculated oocytes, where the protective layer of the oocyte is removed, exhibit short lifetimes (typically less than two weeks from surgical delivery) [7]. Unlike normal cells, oocytes cannot be stored at liquid nitrogen temperatures, and must be harvested from the frogs every time through a surgical procedure. The short time limits the number of experiments that can be performed on a typical oocyte. Since the maintenance of a live *X. laevis* frog specimen requires considerable effort and upkeep, many labs opt for commercial sources for oocytes. Transport logistics and delivery cause the time window for experiments to further shorten their useful time range. Standard housekeeping practices, such as the frequent replacement of oocyte storage solution and housing oocytes at 17 °C are the best methods so far at maintaining the oocytes health for as long as possible but they do not increase the lifespan of the oocyte beyond 7–10 days. Certainly, finding a solution to expand the natural lifespan of *X. laevis* oocytes under lab conditions is highly important. 

We hypothesized that the viability of defolliculated oocytes can be improved by placing them in an environment that is similar to the natural environment of the frog ovary. This natural environment consists of an inner thecal layer composed of blood vessels and an extracellular matrix. Oocytes in the early stages of oogenesis are located within this layer, gradually growing and finally expanding out of the thecal layer and attaching to the inner epithelium of the ovary [8]. Attachment to the inner epithelium provides the oocyte with structural support and access to nutrients. This rationale led us to examine the efficacy of a basement membrane extract (BME) that creates a pseudo-3D environment. While 3D media is commonly used in preclinical oncology to protect cancer cells implanted into animal models by typically using Matrigel [9,10], or more recently, Cultrex [10], explorations into its use as a solution to increase oocyte lifespan have not been explored.

Herein, we placed oocytes in a Cultrex 3D matrix and compared their longevity with oocytes stored under standard conditions for oocyte incubation. To demonstrate the neutrality of the 3D media on transfection, the oocytes were injected with either mRNA for GFP or with cDNA carrying eGFP. Over the course of the study, oocytes were monitored for viability, circularity, membrane stability, and fluorescence intensity as quantitative markers of longevity.

## 2. Materials and Methods

### 2.1. Oocytes

Defolliculated stage V and VI oocytes from *X. laevis* were obtained from EcoCyte Bioscience, Austin, TX, USA. Upon delivery, oocytes were removed from the transport vial and stored in 1× ND96 saline solution (NaCl 96 mM, KCl 20 mM, CaCl_2_·2H_2_0 1.8 mM, MgCl_2_·6H20 1 mM, HEPES 5 mM, pH 7.6) supplemented with sodium pyruvate (2.5 mM) and 1 mL/100 mL penicillin-streptomycin (10,000 units penicillin, 10 mg/mL streptomycin). The oocytes were stored at 17 °C with the storage media changed daily.

A minimum of ten oocytes were used per treatment group and the experiments were repeated several times. The number of oocytes per group was due to the logistics of delivering oocytes from the commercial source with a limited number of oocytes per batch. Each oocyte was stored and handled in an individual chamber to prevent interference with each other. As we observed, oocytes placed in the separate chamber tended to survive longer regardless of the media, and the results were more reproducible. For circularity, fluorescence and microscopy, different groups were used; with each group having at least ten oocytes per group. 

### 2.2. cDNA Preparation

Plasmid pUNIV-EGFP [11] was a gift from Cynthia Czajkowski (#247060, Addgene, Watertown, MA, USA) and was received as a bacterial stab of transfected *Escherichia coli.* The transfected *E. coli* were re-plated on nutrient agar plates with ampicillin and grown at 37 °C overnight. After 12 to 18 hours, the single colonies were transferred into 4 mL of previously frozen liquid broth medium containing ampicillin (final conc. of ampicillin was 100 µg/mL) in a 4 mL cell culture tube. The cultures were then incubated overnight at 37 °C under constant shaking. The bacteria were then collected, and the plasmids were isolated from the bacterial cells using a Spin Miniprep Kit (Qiagen, Ann Arbor, MI, USA). The plasmids were stored on ice for several days prior to the injections into the oocytes. The final concentration of eGFP-cDNA was found to be 200 ng/µL when measured via Nanodrop (ThermoFisher, Waltham, MA, USA). 

### 2.3. GFP mRNA Preparation

GFP mRNA (MRNA11-20, OZ Biosciences, San Diego, CA, USA) was received on dry ice. After thawing, the content was diluted with fresh Milli-Q water and the concentration was measured via Nanodrop to be 120.4 ng/µL. The mRNA was stored at −80 °C.

### 2.4. Cultrex Preparation

Cultrex PathClear Reduced Growth Factor BME (Cultrex RGF BME, 16.1 mg/mL) was obtained from R&D Systems Inc., Minneapolis MN, USA. To prevent potential oocyte membrane rupture and osmosis-induced apoptosis due to incongruent osmolarity between ND96 and Cultrex solution [12], Cultrex was diluted to approximately 5.0 mg/mL by adding 2146.7 µL ND96 storage solution and 1073.3 µL Milli-Q water. The resulting osmolarity of the diluted Cultrex solution was measured to be 180 mOSM/kg (3320 Osmometer, Advanced Instruments Inc., Norwood, MA, USA), which was within the acceptable 175–185 mOSM/kg range to prevent membrane rupture. 

### 2.5. Transfection of X. laevis Oocytes

The oocytes were injected with either 13.5 nL of eGFP cDNA (200 µg/µL) or 46.0 nL GFP mRNA (120.4 ng/µL) using a micro-injector (Nanoliter 2010, World Precision Instruments) [13]. Glass micropipettes were pulled on a micropipette puller (P-1000, Sutter Instrument), using borosilicate glass capillaries (internal diameter (ID) = 0.530 mm, outer diameter (OD) = 1.14 mm, WPI). Prior to eGFP cDNA injection, the oocytes were aligned with the animal pole facing upward (where the nucleus is located), and the injection needle was positioned for the best approximation of direct nuclear injection. For oocytes injected with GFP mRNA, injection sites in the vegetal pole were used. All injections were performed under a standard stereo-dissecting microscope. Following injections, oocytes were placed in individual wells of a 96-well plate containing either ND96 storage solution or diluted Cultrex solution. The well plate was covered and then stored at 17 °C in an incubator for the duration of the experiment.

### 2.6. Fluorescence and Brightfield Imaging

Prior to imaging, oocytes were washed in a petri dish containing ND96 solution for 1 minute. Once the wash step was completed, oocytes were then submerged in ND96 storage solution and placed in a glass dish on the microscope stage (BX51WI, Olympus, Tokyo, Japan). Fluorescent images were taken with a metal halide light source (X-Cite, Excelitas Technologies, Waltham, MA, USA), a FITC 5058A-OMF filtercube (Semrock, IDEX Corp., Lake Forest, IL, USA), NIKON LU Plan Fluor 5X/0.15 objective (Nikon, Tokyo, Japan), thermoelectrically cooled charge-coupled device (CCD) camera (ORCA-ER, Hamamatsu Photonics, Hamamatsu City, Japan) and recorded using HCImageLive software (Hamamatsu Photonics, Hamamatsu City, Japan). Fluorescence intensity was measured in the area corresponding to the oocyte’s vegetal pole region of interest (ROI). The ROI fluorescence corresponded to an average intensity of the similarly sized ROIs. Brightfield imaging was recorded using a halogen lamp attached to the microscope. Images were analyzed for circularity through ImageJ software. Circular shapes with measurement values closer to 1.0 indicate perfect circles while values closer to 0.0 indicate elongated polygons.

### 2.7. Sectioning of Oocytes

Oocytes were pipetted out of the well plate and placed in a disposable base mold (Fisher Scientific). The excess solution was then pipetted out of the base mold to prevent ice crystals from forming in the optimal cutting temperature (OCT) cryotome mounting media. To prevent the oocyte membranes from rupturing due to direct exposure to air, the oocyte was immediately covered in OCT compound completely filling up the mold. The mold was then placed in a −80 °C freezer for 30 minutes to ensure complete freezing and then stored at −20 °C. Oocytes in OCT compound blocks were sectioned at 10 µm using a CM1800 Cryostat (Leica, Wetzlar, Germany). Sections were taken from both the vegetal and animal poles. Sections were then placed on slides and stored at −20 °C. Prior to imaging, the mounting medium Fluoroshield (Sigma-Aldrich, St. Louis, MO, USA) was applied to the sections and coverslips were placed on each slide.

## 3. Results

### 3.1. Cultrex Media Provides Longer Survival

The physical shape of frog oocytes serves as a reliable marker of general oocyte health, with oocytes lacking a pronounced spherical shape and/or severe blebbing (Figure 1) deemed unhealthy or dead. The maintenance of oocyte physical integrity is critically important as deteriorating oocytes can no longer be used for genetic and electrophysiological experiments (i.e., unable to form a reliable giga-ohm seal in patch clamp experiments). 

Oocytes that were placed in the control ND96 solution began rapid deterioration within a few days as can be clearly seen in the survival plot (Figure 2**)**. By the end of the first week, only 20% of defolliculated oocytes stored in ND96 remained healthy and viable for physical manipulation. The visual appearance of the oocytes under a microscope served as a measure of their health and viability. The visual parameters included the stability of the oocyte membrane with no visual membrane blebbing or wrinkling. Oocytes were also judged on their ability to survive limited physical manipulations, such as moving from an incubator to a microscope stage and changing media. This finding falls in line with typical lab estimates of oocyte health once defolliculation has occurred. In contrast, 80% of defolliculated oocytes stored in the Cultrex-supplemented ND96 media remained healthy and viable even after three weeks.

### 3.2. Oocytes Retains Spherical Shape for Longer Time

Aside from overall health, circularity measurements performed on oocytes showed that Cultrex allowed the oocytes to maintain a relatively steady spherical shape over a longer period of time compared to oocytes The ROI fluorescence corresponded in ND96 (Figure 3A). The higher circularity of oocytes placed in the Cultrex-ND96 media suggested that the hydrogels are able to provide stable environments to support oocyte structure while helping to maintain its functional integrity. Oocytes incubated in the presence of Cultrex exhibited maximal GFP expression for a period of three weeks and did not show any formation of blebbing, an unfortunate by-product of many experiments that follow microinjection of *X. laevis* oocytes. As illustrated in Figure 3B,C, three-week-old oocytes were able to express fluorescent proteins derived from either cDNA or mRNA and maintain the healthy circular shape. Taken together, the survival plot and the corresponding circularity measurements indicate the efficacy of Cultrex as a potential additive for increasing the lifespan of oocytes. 

### 3.3. Cultrex Maintains Oocyte Membrane Integrity

To demonstrate the advantage of Cultrex-ND96 vs a standard ND96 media we looked at the integrity of the membrane. Figure 4 shows an oocyte sectioned on the same day of the delivery from the vendor. A thick membrane that surrounds the oocyte keeps the shape of the oocyte circular, as shown in Figure 4A. The oocytes sectioned after 3 days of being in ND96 revealed a large number of black vesicles separating from the cell body and membrane blebs (Figure 4B). These blebs are an indicator of poor oocyte health and are a precursor to membrane fracture and failure. Oocytes stored in Cultrex-ND96 media were able to maintain their membrane integrity as demonstrated in Figure 4C. After 12 days most of the oocytes (eight of the ten oocytes) stored in Cultrex-ND96 presented intact membranes and no noticeable blebbing. In contrast, only two of the ten oocytes stored in the ND96 media had survived by day 12. Both of these two oocytes in ND96 media presented disintegrated membrane fractures (Figure 4D). 

### 3.4. Cultrex Does Not Interfere with mRNA or cDNA Derived Protein Expression

While the increased longevity of oocytes in Cultrex media is a useful improvement, longevity alone does not imply that the oocytes will necessarily be fully functional. *X. laevis* oocytes are well known as an established platform for the expression of a wide variety of heterologous proteins [14,15]. To test whether Cultrex media interferes with the native cellular mechanisms we investigated the dynamic expression of exogenous proteins. We injected oocytes with either GFP mRNA or transfected them with the eGFP plasmid pUNIV-EGFP [13]. We relied on standard methodology for the injection of foreign mRNA into the oocyte cytoplasm or the transfection of foreign cDNA plasmids directly injected into the oocyte nucleus. Once the injections were complete, oocytes were placed in either control ND96 saline solution or Cultrex-ND96 matrix. The expression of the fluorescent proteins was judged via the fluorescence intensity of the oocyte’s vegetal pole due to the melanin pigmentation on the oocyte’s animal pole. The images of the oocytes were recorded every 2 to 3 days over a period of 3 weeks until the death of the oocyte. Only visibly viable oocytes were recorded. Over the course of the experiment, oocytes were discarded if the fluorescence was significantly decreased, or the oocytes’ shapes were visibly disrupted/ruptured. 

During the first several days after the mRNA injection the oocytes placed in both types of media showed comparable levels of fluorescence expression (Figure 5) suggesting that Cultrex has little effect on the protein expression during the initial incubation post-injection. After the first 10 days, more dramatic changes were observed. Oocytes incubated with ND96 alone witnessed a sharp decline in the fluorescence signal compared to oocytes incubated in Cultrex-ND96 media. Oocytes in the Cultrex-ND96 media continued to exhibit an increase in fluorescent intensity for an additional 8 days. This increase in protein expression was in line with the survival results (Figure 2). Only a few oocytes incubated in ND96 survived after day 10 (less than 10–20%), while the number of surviving oocytes incubated in Cultrex-ND96 remained high (80%). Both cDNA-injected oocytes (Figure 5B) and mRNA-injected oocytes (Figure 5A) in Cultrex-ND96 were able to express fluorescence proteins until days 17–21. In contrast, oocytes placed in under control ND96 solution exhibited either a plateau in GFP expression (Figure 5B) or a decrease in expression (Figure 5A) until death by the end of the second week. 

## 4. Discussion

One of the important questions is to understand the mechanism by which Cultrex prolongs the survival of *X. laevis* oocytes. While our initial 3D media hypothesis centers on the natural environment surrounding in situ-housed oocytes in the ovary, the answers to the mechanism may be found in the contents of Cultrex solution itself as well as the mechanism of oocyte degradation. 

Cultrex, like its older sibling Matrigel, is a basement membrane extract (BME) purified from Engelbreth -Holm -Swarm tumor. Basement membranes are thin layers of a highly biologically active extracellular matrix that form an interface between endothelial, epithelial, fat, muscle, or neuronal cells and their adjacent stroma and play an essential role in tissue support, growth, and homeostasis. The composition of the extract is highly complex, with major components that include laminin, collagen IV, entactin, and heparan sulfate proteoglycans, which are crucial in maintaining membrane support, cell organization, blood vessel and nerve growth guidance, and forming key mechanical barriers to the spread of microbes and cancer cells. The presence of these “support molecules” apparently plays a crucial role in maintaining the shape and health of the defolliculated oocyte over a longer period when compared to the standard storage solution. Added to living tissue, the extract polymerizes and provides a natural extracellular matrix hydrogel forming a new reconstituted basement membrane. This property explains the traditional clinical use of BME in tissue engineering to improve graft survival and repair damaged tissues and preclinically increase tumor growth from implanted cells in rodents [16].

The structure of the *Xenopus* oocytes is composed of the two poles (animal and vegetal), enclosed in three layers that includes a plasma membrane rich with receptors and ion channels, a fibrous vitelline membrane that performs several functions including protection, and a follicular cell layer. Defolliculation leaves oocytes with an exposed vitelline membrane. Similar to zona pellucida in mammalian oocytes, vitelline is composed of a few glycoproteins of the ZP family [17] that form a tight fibril coat around the oocyte [18]. The sugar chains from ZP proteins play an important role in species-specific interactions with sperm and help block polyspermy. 

ZP proteins in *Xenopus* oocytes are likely to be synthesized by the oocyte itself and not by the follicular cells as demonstrated through a number of studies involving mouse oocytes [19]. The rate of synthesis of ZP proteins in mouse oocytes is the highest in the growing stage and reaches its maximum thickness of the vitelline membrane right before maturation. Many species, including *X. laevis*, show rapid oocyte maturation if follicular cells are removed [20]. Mature oocytes stop producing new ZP proteins and therefore cannot replenish the loss of ZP proteins due to the damage to the coat. Such damage can occur through a variety of mechanisms (e.g., pH extremes, heat, proteinases, reducing agents, physical contact) that disrupt covalent or noncovalent interactions between the ZP proteins within the fibril shell [19]. The absence of the repair mechanisms might explain the lack of vitelline stability after the follicular cell layer removal. The frequent replacement of the media minimizes the deterioration effect of these factors, but only up to a certain extent. 

One of the explanations for the improved survival of defolliculated oocytes in Cultrex is that Cultrex proteins and glycoproteins form a gel-like capsule around the oocyte, preventing ZP proteins from degradation and keeping the vitelline membrane intact. It is also possible that glycoproteins such as laminin or heparan sulfate from Cultrex adhere to the surface of the exposed plasma membrane after the vitelline membrane deteriorates, effectively sealing the damage. Many of these glycoproteins are exogenously expressed in oocytes in a variety of species and play important roles at different stages of oocyte development and fertilization [21,22]. However, finding the role of each glycoprotein will require additional effort.

The role of yolk platelets in oocyte health needs also to be further investigated. In healthy oocytes the yolk platelets fuse together into larger and larger vesicles taking up to 50% of the volume of the oocyte and containing 80% of all protein content [23]. As the oocytes aged and began to deteriorate, the large yolk vesicles appeared to morph into fractured vesicles as illustrated in Figure 4D. These fractured vesicles may be the yolk of the oocyte breaking down into individual yolk platelets. In the older oocytes not stored in Cultrex, the yolk platelets may be breaking down into smaller and smaller vesicles reverting to their smaller size from when the oocyte was earlier in development. Comparing the day 12 Cultrex and ND96 oocytes, you can see the difference in yolk vesicle and yolk platelet distribution (Figure 4C and 4D). In the ND96 stored oocyte the platelets are freely moving away from the larger yolk vesicles in the oocyte and are not tightly packed. In the Cultrex stored oocyte the yolk vesicles appear larger and more ordered than in the ND96-stored oocyte. The reason these yolk vesicles could still be tightly packed is that the stronger membrane integrity of the Cultrex oocyte better maintains the shape and overall function of the oocyte.

## 5. Conclusions

Cultrex 3D media provides a new method for extending the laboratory viability of defolliculated *X. laevis* oocytes. In addition to the increased longevity of the oocytes, we established that Cultrex media did not interfere with the expression of exogenous protein levels in the oocyte when compared to standard conditions. Thus, coupling this maximal expression with longer physical vigor will allow researchers an extended experimental time window to perform previously limited longitudinal studies. It is expected that the proposed technique will greatly reduce the number of frogs used for oocyte harvesting. This reduction is in line with the 3R principles that refer to the replacement, reduction, and refinement of animals used in research, teaching and testing [24]. Our future work will be focused on answering some of the important questions about the prolonged life of these oocytes. What is the molecular mechanism of lifespan prolongation? Does the treatment change the plasma membrane and affect receptors or ion channels? Is this type of mechanism common to other species, including humans? Given the striking homology of ZP proteins between *X. laevis* and human oocytes (88% for ZP2, 85% for ZP3 and 85% for ZP4), can this approach be also used in medicine i.e., to improve fertilization? In that regard, Matrigel has been recently reported as a promising 3D media to derive fully fertilizable oocytes from immature follicles [25]. The use of novel bio-inspired approaches with non-toxic materials can potentially reduce the deleterious effects observed with conventional in vitro fertilization, resulting in improving the viability and functionality of human oocytes.

## Figures and Tables

**Figure 1 membranes-12-00754-f001:**
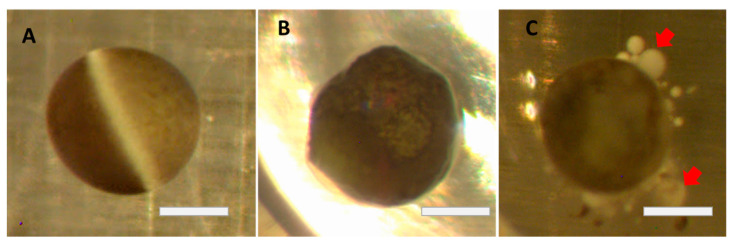
Changes in oocytes after defolliculation (brightfield images): (**A**) Perfect, healthy oocyte. (**B**) Change of circularity due to aging. (**C**) Strong blebbing (red arrows) minutes before death. Bars = 0.5 mm.

**Figure 2 membranes-12-00754-f002:**
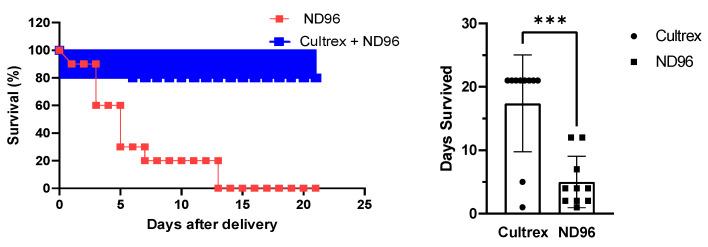
Effect of Cultrex on the survival of oocytes stored in ND96 (control) and Cultrex-ND96 solution. (n = 10 for each media). (**Left**) Percent Survival. (**Right**) Days survived. *** Unpaired t-test, *p* < 0.001.

**Figure 3 membranes-12-00754-f003:**
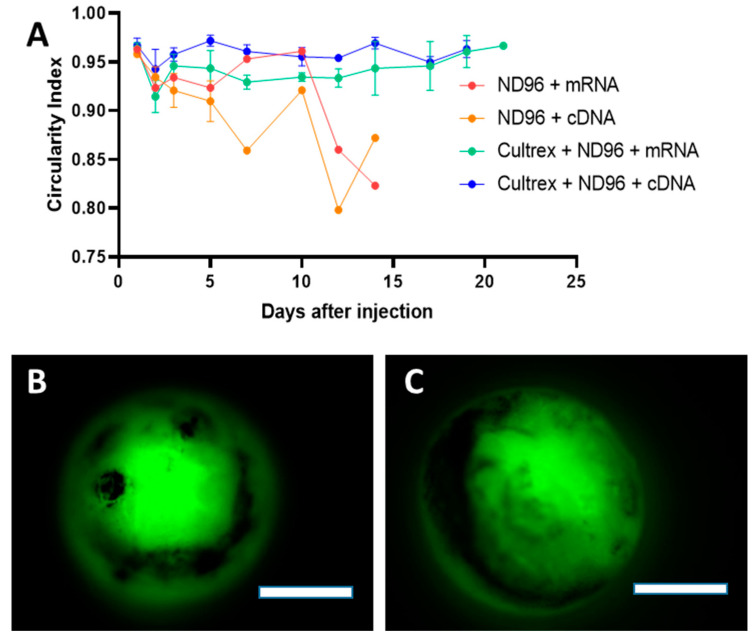
Circularity of oocytes incubated in control ND96 or Cultrex-ND96 media. (**A**) Circularity index of survived oocytes. Values closer to 1.0 indicate perfect circles while values closer to 0 indicate the complete loss of circularity. Loss of standard deviation bars for some data points is the result of survival at those time points being n = 1. (**B**) Fluorescent images from an oocyte expressing fluorescent proteins from cDNA at Day 21 stored in Cultrex-ND96. (**C**) Fluorescent images from an oocyte expressing fluorescent proteins from mRNA at Day 21 stored in Cultrex-ND96 media. Bars = 0.5 mm.

**Figure 4 membranes-12-00754-f004:**
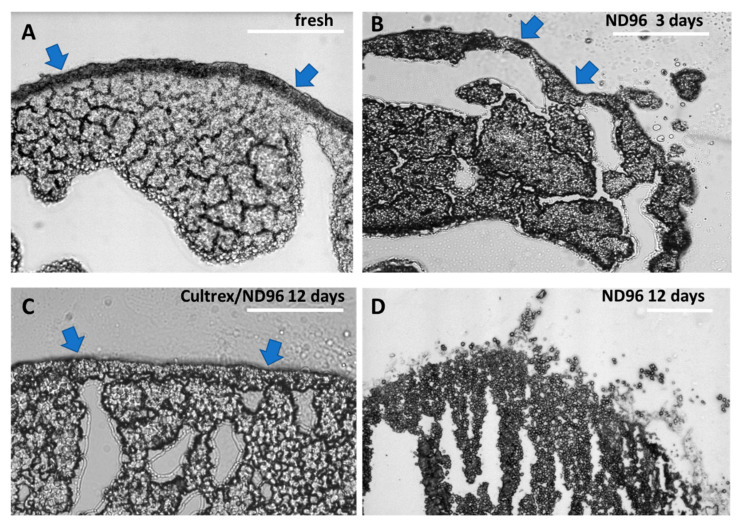
Oocytes sections using brightfield microscopy. (**A**) Day 1 fresh oocyte, (**B**) day 3 oocyte exhibiting moderate blebbing in ND96 storage solution. (**C**) Day 12 oocyte stored in Cultrex-ND96. (**D**) Day 12 oocyte stored in ND96. Olympus BX51, 20× objective. Blue arrows indicate oocyte membrane. Bars = 0.15 mm.

**Figure 5 membranes-12-00754-f005:**
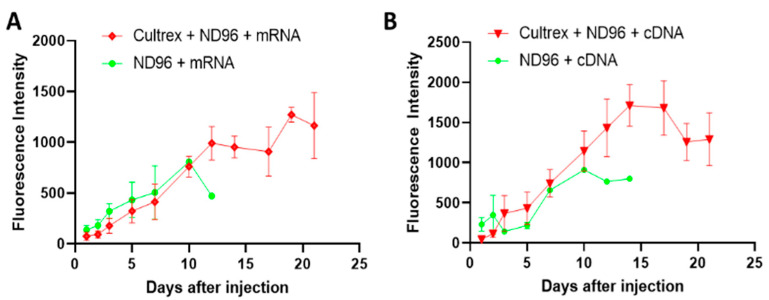
Expression of the fluorescent proteins after the injection of mRNA or cDNA into the oocytes. (**A**) Oocytes injected with GFP mRNA in Cultrex-ND96 vs. control (ND96) media. (**B**) Oocytes injected with cDNA plasmids for eGFP in Cultrex-ND96 vs. control (ND96) media. Non-injected oocytes did not express GFP and therefore did not produce any fluorescence beyond the background (not shown).

## Data Availability

Data is contained within the article. The raw images can be also available per request.

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
