# Peer review of "3D Media Stabilizes Membrane and Prolongs Lifespan of Defolliculated Xenopus laevis Oocytes"

_membranes, 2022, doi:10.3390/membranes12080754_

Round 1
Reviewer 1 Report
Overall this is an interesting study and the results are very clear although the language needs tidying up. I have made some specific suggestions that I hope will improve the manuscript below.
Specific comments:
One thing I felt was missing from the manuscript is that this discovery will impact the 3Rs in terms of reduction. Fewer animals will be required to do the same number of experiments if this technique is adopted. I strongly suggest that this is made explicitly clear.
Line 138 – how was an oocyte that “remained healthy and viable for physical manipulation” defined in the study?
Line 141 – In Fig 2 the cultrex cultured oocytes seem to be stable until the end of the experiment, is how long they are stable for actually known?
Line 160 – In fig 3 would not parts B and C better fit in fig 5?
Reviewer 2 Report
Manuscript: “3D Media Stabilizes Membrane and Prolongs Lifespan of De-folliculated Xenopus laevis Oocytes”
Manuscript Summary:
The manuscript describes a 3D Cultrex-based storage for defolliculated Xenopus laevis oocytes. Using this system, survival and integrity of oocytes were longer than in the control standard conditions (ND96 saline solution) and the oocytes remained usable for transfection of mRNA or cDNA.
Major Concerns:
The number of oocytes is very low and unclear. In Figure 2 it is stated that 10 oocytes were used for each medium, which is low. Please state the number of oocytes used for circularity, fluorescence and electron microscopy. Please also explain whether different oocytes were used for different analysis or if some oocytes were used for more than one analysis.
Minor Concerns:
- Acronyms need to be explained
- The sentence “(Error! Reference source not found.)” appears many times in the text. I guess reference or links to hypertext should be added (lines 127, 136, 149, 155, 202, 208, 211, 212, 214).
- Please removed the unpaired parenthesis at line 170
- Please modify Figure 4. Each panel (A, B, C, D) has a sentence written on it, but this is not well visible for B, C and D.
- Fluorescence images were recorded using HCImageLive software (Hamamatsu) – line 110. How was the fluorescence measured?
- Lines 198-199: Please show the data about viable oocytes. Were there also control non-injected oocytes?
Round 2
Reviewer 2 Report
The authors answered thoroughly to my previous comments and revised the manuscript accordingly.
Author Response
We thank the Reviewer for the suggestion and added the scale bars to both Figures as suggested